# Structural insights into NDH-1 mediated cyclic electron transfer

Chunli Zhang[1,2,3,10], Jin Shuai[4,5,10], Zhaoxing Ran [6,10], Jiaohong Zhao[6], Zhenfang Wu[1,2], Rijing Liao [1,2], Jian Wu [1,2,7✉], Weimin Ma [6,8✉] & Ming Lei [1,2,9✉]

NDH-1 is a key component of the cyclic-electron-transfer around photosystem I (PSI CET) pathway, an important antioxidant mechanism for efficient photosynthesis. Here, we report a 3.2-Å-resolution cryo-EM structure of the ferredoxin (Fd)-NDH-1L complex from the cyanobacterium *Thermosynechococcus elongatus*. The structure reveals three β-carotene and fifteen lipid molecules in the membrane arm of NDH-1L. Regulatory oxygenic photosynthesis-specific (OPS) subunits NdhV, NdhS and NdhO are close to the Fd-binding site whilst NdhL is adjacent to the plastoquinone (PQ) cavity, and they play different roles in PSI CET under high-light stress. NdhV assists in the binding of Fd to NDH-1L and accelerates PSI CET in response to short-term high-light exposure. In contrast, prolonged high-light irradiation switches on the expression and assembly of the NDH-1MS complex, which likely contains no NdhO to further accelerate PSI CET and reduce ROS production. We propose that this hierarchical mechanism is necessary for the survival of cyanobacteria in an aerobic environment.

[1] Ninth People's Hospital, Shanghai Jiao Tong University School of Medicine, 200125 Shanghai, China. [2] Shanghai Institute of Precision Medicine, 200125 Shanghai, China. [3] Phil River Technology, 100042 Beijing, China. [4] State Key Laboratory of Molecular Biology, CAS Center for Excellence in Molecular Cell Science, Shanghai Institute of Biochemistry and Cell Biology, Chinese Academy of Sciences (CAS), 200031 Shanghai, China. [5] University of Chinese Academy of Sciences, CAS, 200031 Shanghai, China. [6] College of Life Sciences, Shanghai Normal University, 200234 Shanghai, China. [7] Shanghai Key Laboratory of Translational Medicine on Ear and Nose diseases, 200125 Shanghai, China. [8] Shanghai Key laboratory of Plant Molecular Sciences, College of Life Sciences, Shanghai Normal University, 200234 Shanghai, China. [9] Key laboratory of Cell Differentiation and Apoptosis of Chinese Ministry of Education, Shanghai Jiao Tong University School of Medicine, 200025 Shanghai, China. [10]These authors contributed equally: Chunli Zhang, Jin Shuai, Zhaoxing Ran ✉email: wujian@shsmu.edu.cn; wma@shnu.edu.cn; leim@shsmu.edu.cn

Cyanobacteria are considered to be the first to provide oxygen by photosynthesis necessary for starting the evolution of complex organisms on Earth[1,2]. However, the rise of oxygen on Earth inevitably resulted in the production of reactive oxygen species (ROS) in cells, especially under environmental stresses. In order to adapt to the environment with ROS that causes damages in cells[3], cyanobacteria have to develop new antioxidant mechanisms to reduce ROS production or scavenge ROS. Among them, cyclic electron transfer around photosystem I (PSI CET) is an important mechanism that balances the ATP/NADPH ratio required for the Calvin–Benson cycle and reduces the ROS production[4,5]. The NDH-1 complex is a key component of this pathway in most photosynthetic organisms and its inactivation leads to growth defects and even cell death under environmental stresses[6,7].

Four types of NDH-1 (NDH-1L, NDH-1L', NDH-1MS, and NDH-1MS') have been identified in cyanobacteria (Fig. 1a)[8–10], and all of them are involved in PSI CET[11]. They all share the common NDH-1M module that contains four oxygenic photosynthesis-specific (OPS) regulatory subunits NdhL, NdhO, NdhS, and NdhV in addition to 11 structural components NdhA–NdhC, NdhE, NdhG–NdhK, NdhM, and NdhN (Fig. 1a)[8,9,12]. To carry out the PSI CET function, the common NDH-1M module has to associate selectively with one of the four variable modules that contain different NdhD and NdhF subunits (Fig. 1a)[8,10]. Two recently published structures of NDH-1 revealed that subunit NdhD mediates the direct interaction with the NDH-1M module and that one β-carotene molecule is involved in stabilizing their association[13,14]. It is noteworthy that the expression pattern of four variable modules changes and makes different contributions to PSI CET when cyanobacteria are exposed to different environmental stresses[15–17].

Previous studies strongly suggest that, together with Fd, the NDH-1M module constitutes a key portion of the PSI CET pathway—electrons donated by photoreduced Fd are transferred to PQ via the chain of three conserved [4Fe-4S] clusters[14,18], corresponding to previously identified N6a, N6b, and N2 clusters in respiratory complex I[19] (Fd-N6a-N6b-N2-PQ). In the NDH-1M module, the 11 structural subunits appear to play an important role in stabilizing the complex for efficient electron transfer, whereas the four OPS regulatory subunits (NdhL, NdhO, NdhS, and NdhV) are involved in controlling the flow rate of electron transfer from Fd to NDH-1M. The flow rate of electron transfer can be quickly boosted during the exposure of cyanobacteria to environmental stresses[20]; however, the regulatory roles of the four OPS subunits in this process still remain poorly understood.

Here we report the cryo-electron microscopic (cryo-EM) structure of entire NDH-1L complex with all 19 subunits (including NdhV) in complex with Fd and reveal the structural basis of the electron transfer chain from Fd to NDH-1L. Our functional data further unveil the regulatory roles of OPS subunits during the exposure of cyanobacteria to high-light stress.

## Results

**Overall structure of NDH-1L in complex with Fd.** We purified the endogenous NDH-1L from the thermophilic cyanobacterium

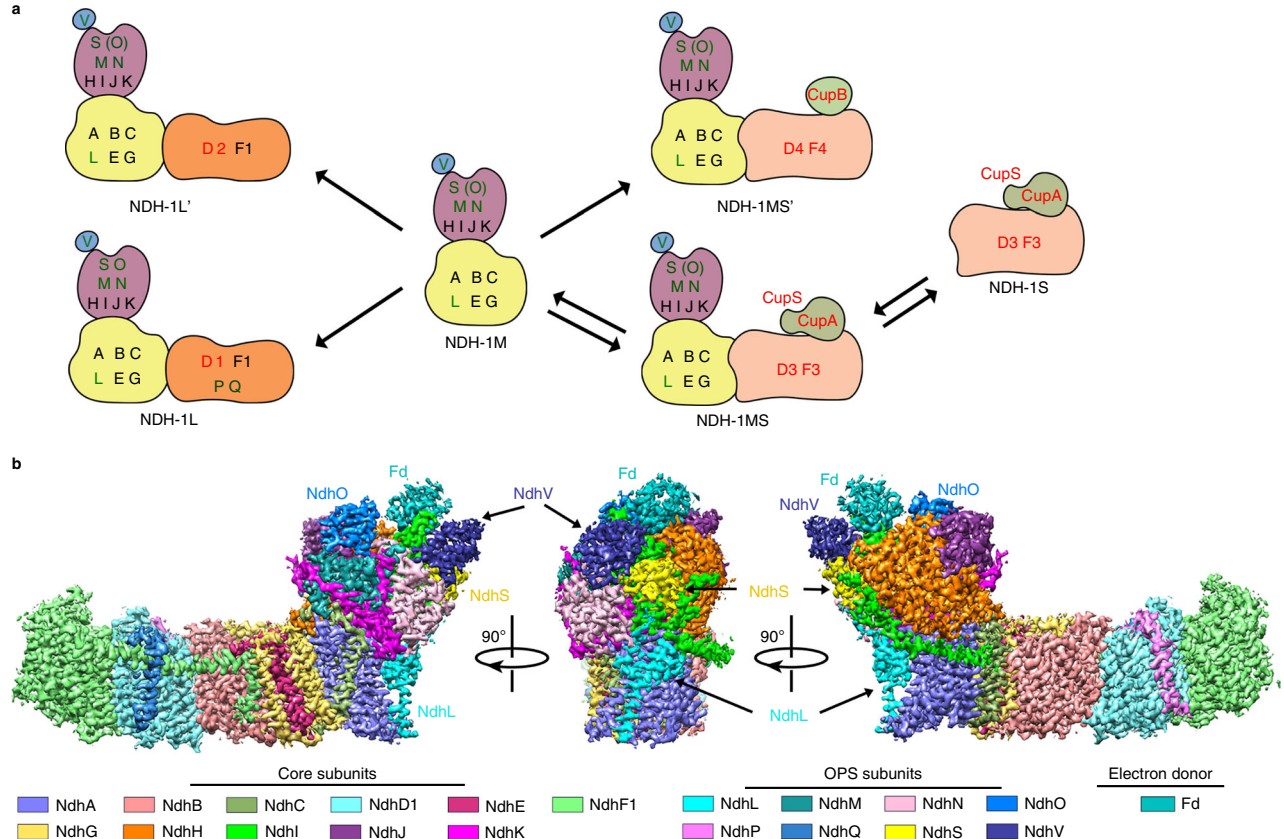

**Fig. 1 Cryo-EM structure of the Fd-NDH-1L complex. a** Subunit compositions of four types of cyanobacterial NDH-1 complexes, all of which share a common NDH-1M module. Subunits are labeled and colored; core subunits in black, OPS subunits in green and isoform-specific subunits in red. NdhV is drawn as a separate subunit to emphasize that it transiently associates with the NDH-1 complexes. **b** Three views of the overall cryo-EM reconstruction of the Fd-NDH-1L complex. The cryo-EM density map is segmented by subunits of NDH-1L. Subunits of NDH-1L and Fd are color-coded and the scheme is shown below.

*Thermosynechococcus elongatus* BP-1 using a native polyhistidine tag on subunit NdhF1 (Supplementary Fig. 1a). Mass spectrometric analysis suggested that the amount of NdhV in the purified complex appeared to be substantially less than that of NdhS, which has a similar molecular weight as NdhV (Supplementary Table 1). This observation is in accordance with previous data that NdhV is a transiently associated subunit of NDH-1[21]. We determined the cryo-EM structure of the NDH-1L complex at 3.6-Å resolution, which indeed lacks the NdhV subunit (Supplementary Figs. 1 and 2 and Supplementary Table 2). Hereafter, we will refer to this complex as NDH-1LΔV.

We next recombinantly expressed and purified NdhV and Fd individually and mixed them with NDH-1LΔV to reconstitute the Fd-NDH-1L complex (Supplementary Fig. 3a, b). Single-particle cryo-EM analysis resulted in a well-defined electron density map at an overall resolution of 3.2 Å (Fig. 1b, Supplementary Figs. 3–5, and Supplementary Table 2). We built an atomic structure of the entire NDH-1L complex with all 19 subunits in complex with Fd (Fig. 1b).

Consistent with the recently reported structures of NDH-1[13,14], 11 core subunits NdhA–NdhK are spatially organized in the canonical L-shaped architecture (Fig. 1b)[22]. Fd binds to the apex of the peripheral arm in a shallow pocket formed by NdhI, NdhH, NdhV, and NdhO (Fig. 1b). The Fd-NDH-1L complex structure reveals four putative proton translocation pathways as well as the arrangement of the redox centers alongside the complete electron transfer pathway—Fd-N6a-N6b-N2-PQ (Supplementary Fig. 6). Similar to respiratory complex I structures, three core subunits, NdhH, NdhK, and NdhA, encircle a putative PQ-binding cavity (Supplementary Fig. 7a)[22]. Notably, at the proximal end of this cavity, the β1–β2 loop of NdhH is well ordered and its conformation closely resembles that in the active state of respiratory complex I structures (Supplementary Fig. 7a, b)[23–26], suggesting that the structure of the Fd-NDH-1L complex captures an active conformation of NDH-1L.

**Cofactors in the membrane arm of the NDH-1L complex.** The EM density map unveiled the existence of three β-carotene molecules (Fig. 2a and Supplementary Fig. 5). The first hint of the chemical identity of β-carotene was from the yellow color of the purified NDH-1LΔV complex revealed by the clear native-polyacrylamide gel electrophoresis (CN-PAGE) analysis (Supplementary Fig. 8a). Absorption spectrum of the excised yellow band from the gel showed the characteristic absorption peaks of β-carotene, further confirming the presence of β-carotene in the purified NDH-1LΔV complex from *T. elongatus* thylakoid membranes (Supplementary Fig. 8b). It is noteworthy that one of the β-carotene densities was also observed in the recently reported NDH-1 structures[13,14]. β-Carotene has been widely found in photosynthetic membrane–protein complexes[27,28]. Our structural and biochemical analyses demonstrated that, except for β-carotene, there are no other pigment molecules in the NDH-1L complex, suggesting that the β-carotene molecules found in NDH-1L are unlikely to be involved in antioxidant mechanism that quenches singlet oxygen from chlorophyll molecules, as proposed in other photosynthetic membrane–protein complexes[29].

In addition to β-carotene, the EM density map also allowed identification of 15 tightly bound endogenous lipid molecules, including 9 dipalmitoylphosphatidylglycerol (LHG), 4 sulfoquinovosyldiacylglycerol (SQD), and 2 digalactosyldiacylglycerol (DGD) (Fig. 2a and Supplementary Fig. 5). Mass spectrometric analysis showed that the observed molecular weights of the co-purified lipids are identical to the calculated values of these thylakoid membrane-specific lipids[30], unequivocally confirming their chemical identities (Supplementary Fig. 8c). Given that we purified the endogenous NDH-1LΔV complex from *T. elongatus* thylakoid

membranes with a mild detergent (digitonin), the lipids we found in the NDH-1L structure are very likely physiologically relevant and reflect their actual binding positions in NDH-1L in vivo, suggesting that lipids in NDH-1L may play important roles as lipids found in other photosynthesis complexes of cyanobacteria[30].

One salient feature of the cofactors found in the NDH-1L complex is that they all fit into the cavities among protein subunits in the membrane arm (Fig. 2a). In particular, there are two β-carotene and four lipid molecules at the interfaces between NdhD1 and its adjacent subunits (Fig. 2b, c). At one side of the membrane arm, one SQD molecule fits into an enclosed hydrophobic hole between NdhD1 and NdhF1 (Fig. 2b, Supplementary Fig. 9a). A β-carotene tightly wraps around half of a transmembrane helix of NdhD1 with one trimethylcyclohexene group contacting the tail of the SQD molecule and the other trimethylcyclohexene group sticking into a hydrophobic cavity between NdhD1 and NdhP (Fig. 2b, Supplementary Fig. 9b). This β-carotene is almost completely covered by two LHG molecules from outside of the transmembrane arm (Fig. 2b, Supplementary Fig. 9c). At the opposite side of membrane arm, another SQD molecule contacts both NdhD1 and NdhF1 in the connecting region between the two subunits and is covered by the long helix of NdhF1 (Fig. 2c, Supplementary Fig. 9a). A β-carotene lies on a flat hydrophobic surface and interacts with NdhD1, NdhF1, and NdhQ (Fig. 2c, Supplementary Fig. 9d). Collectively, these cofactors help keep NdhD1, NdhF1, NdhP, and NdhQ together as an NDH-1L-specific module and stabilize its connection to the common NDH-1M module (Figs. 1a and 2b, c).

Another area with concentrated cofactors is at the heel of the L-shaped complex (Fig. 2d, e). One SQD snugly fills the cavity formed by NdhA, NdhL, NdhK, and NdhN, stabilizing the connection between the transmembrane and peripheral arms (Fig. 2d). Strikingly, a continuous cofactor belt, composed of two DGD, one LHG, and one β-carotene, almost wraps around half of the NdhA surface from the luminal to the stromal sides of the membrane (Fig. 2d, e). At one end, two DGD molecules intermingle together and fit into a large hydrophobic cavity in NdhA with their head groups toward the luminal side and stabilized by an extensive network of hydrogen-bonding interactions with NdhA (Fig. 2e). Notably, both DGD lipids are capped by NdhL at the entrance of the putative PQ-binding cavity formed by NdhA, NdhK, and NdhH (Fig. 2d). It is plausible that NdhL together with these lipid molecules play an important role in stabilizing the PQ-binding cavity for efficient electron transfer from Fd. At the other end of the cofactor belt, one LHG molecule fits into a hydrophobic cavity formed by one side of NdhA and the N-terminal long helix of NdhI, with its head group contacting NdhH at the stromal sides of the membrane (Fig. 2e, Supplementary Fig. 9e). Notably, in the middle of the cofactor belt, a β-carotene snugly sits in a highly hydrophobic groove and mediates extensive interactions with numerous aromatic and hydrophobic residues from NdhA, NdhL, and NdhI, connecting the tails of DGD and LHG molecules by its trimethylcyclohexene groups at both ends (Fig. 2e, Supplementary Fig. 9e). Taken together, based on their locations in the NDH-1L complex structure, it appears plausible that these hydrophobic β-carotene and lipid molecules play an important role in stabilizing the NDH-1L complex for efficient electron transfer and proton translocation.

**The Fd-binding site.** The peripheral arm of NDH-1L adopts a cylinder-shaped architecture organized by four conserved core subunits (NdhH–NdhK) and two OPS structural subunits (NdhM and NdhN) (Fig. 3a). The central axis of the cylinder is the redox chain of three [4Fe-4S] clusters composed of distal and medial clusters N6a and N6b coordinated in NdhI and terminal cluster

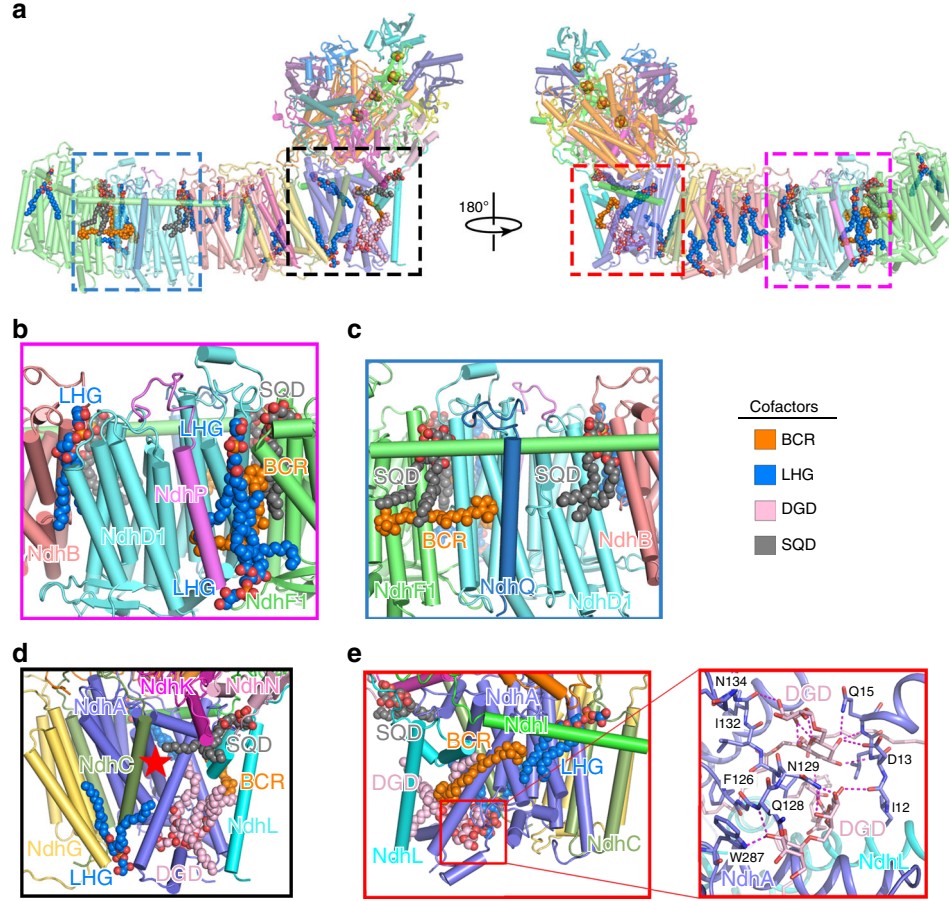

**Fig. 2 Hydrophobic cofactors identified in the Fd-NDH-1L complex structure. a** Two side views of the Fd-NDH-1L complex with cofactors shown as spheres, colored as in Fig. 1b. Cofactors are color-coded and the scheme is shown below. BCR β-carotene, LHG dipalmitoylphosphatidylglycerol, SQD sulfoquinovosyldiacylglycerol, DGD digalactosyldiacylglycerol. **b, c** Close-up views of cofactor locations at the interfaces between NdhD1 and NdhP, NdhF1, and NdhB (**b**) and between NdhD1 and NdhQ, NdhF1, and NdhB (**c**). **d, e** Close-up views of cofactor locations in the heel of the complex around NdhA and NdhL. Red star denotes the entrance to the PQ-binding cavity (**d**). Interactions between DGD and NdhA/NdhL in the box region are enlarged in the inset shown below (**e**). Dashed magenta lines denote the intermolecular hydrogen-bonding interactions.

N2 in NdhK (Fig. 3a). OPS regulatory subunits NdhS, NdhV, and NdhO all fold into small compact structures, binding to the apex of the peripheral arm (Fig. 3a). NdhS stably sits in a groove of NdhI, holding NdhV to attach to and buttress NdhI for Fd binding and electron transfer (Fig. 3a). On the opposite side of NdhI, NdhO is secured on the surface of the peripheral arm through interactions with NdhJ, NdhK, and NdhN (Fig. 3a).

On top of the peripheral arm, Fd binds into a concaved, highly positively charged surface formed by NdhI and NdhH as well as NdhV and NdhO (Fig. 3a, b). The side chain positive charges of Lys86, Lys89, and Lys91 of NdhI and Lys315 and Lys321 of NdhH contact the negative patches of Fd through electrostatic interactions (Fig. 3c). In addition, two positive clusters of NdhV and NdhO (Arg122 and Lys123 of NdhV and Lys4 and Lys5 of NdhO) extend the Fd–NDH-1L interface from opposite sides (Fig. 3c). In contrast, an electrostatic interaction of NdhS with Fd is not observed in the Fd-NDH-1L complex structure (Fig. 3c). This structural information implies that NdhV and NdhO but not NdhS are directly involved in association of Fd to the NDH-1L complex and electron transfer.

Structural comparison reveals that the Fd-binding pocket of NDH-1L highly resembles that of PSI, which is also formed by four subunits—PsaC, PsaA, PsaD, and PsaE (Fig. 3d). In particular, Fd binds to NdhI in NDH-1L in the same manner as Fd to PsaC in PSI (Fig. 3d). Consequently, the distance and

angular relationships between the [2Fe-2S] cluster in Fd and [4Fe-4S] clusters in NDH-1L or PSI are almost identical (Fig. 3d), explaining how Fd shuttles between NDH-1 and PSI for efficient electron transfer[14,31]. Notably, the electrostatic surface potentials of the Fd-binding pockets are quite different in NDH-1L and PSI; the Fd-binding surface is highly positive in NDH-1L but close to neutral in PSI (Fig. 3e). Such difference is in accordance with the opposite directions of electron flow in the two photosynthetic membrane–protein complexes (Fig. 3d, e).

**Regulatory roles of OPS subunits in response to high light.** In response to high-light stress, cyanobacteria rapidly increase the NDH-1-dependent PSI CET (NDH-CET) within several minutes (Fig. 4a)[20]. Such response plays a key role in preventing accumulation of excess electrons at the acceptor side of PSI and thus protecting cyanobacteria against ROS damage[32,33]. The Fd-NDH-1L complex structure reveals that all four OPS regulatory subunits are in close vicinity to the electron transport pathway in NDH-1L–NdhL adjacent to the electron acceptor PQ cavity while NdhV, NdhS, and NdhO are at the electron donor Fd-binding site (Supplementary Fig. 6).

Previous in vivo chlorophyll fluorescence studies showed that under growth-light conditions deletion of NdhV or NdhS resulted in decreased NDH-CET activity, whereas removal of NdhO led to increased activity, suggesting that they play different roles in

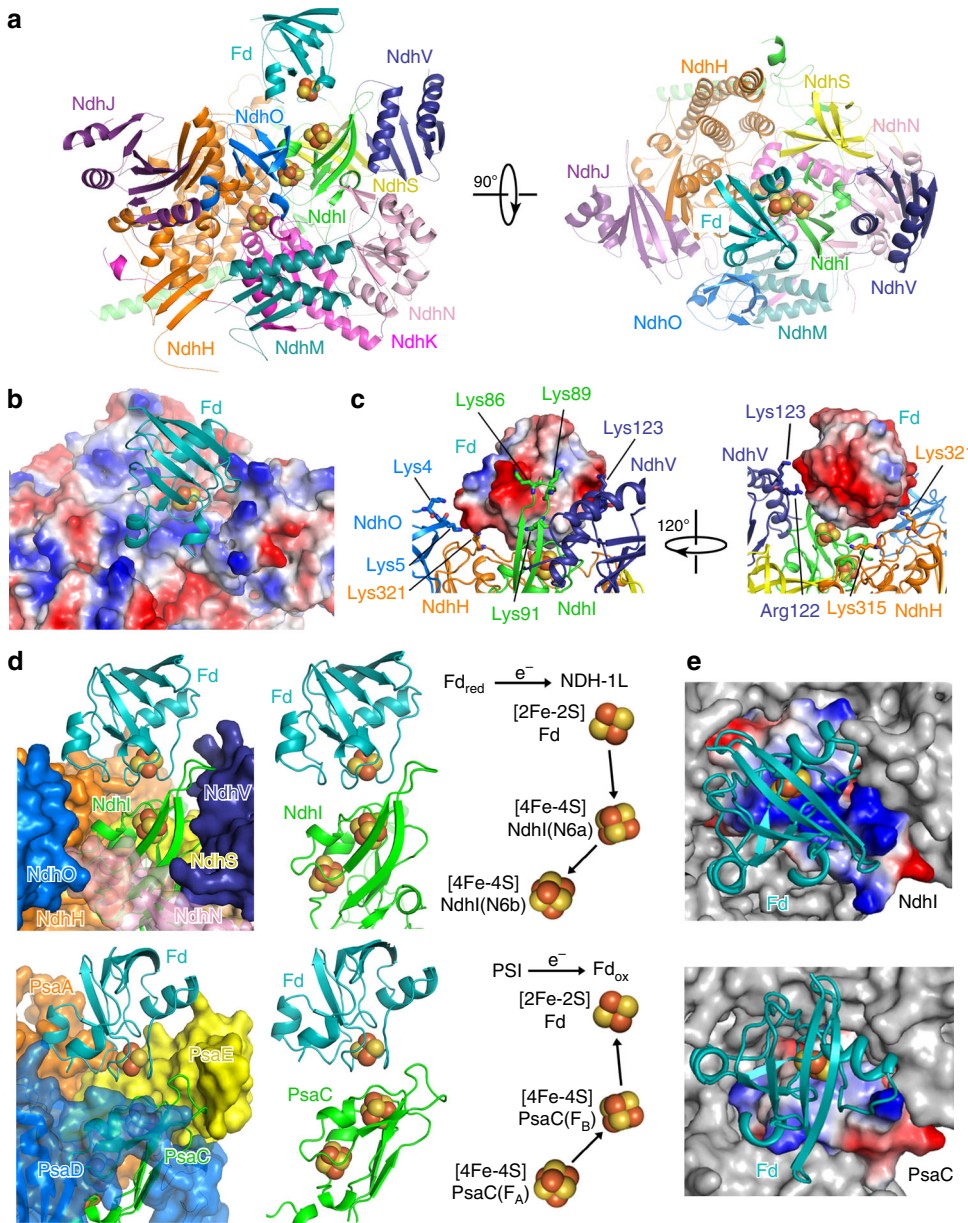

**Fig. 3 The Fd-binding pocket of NDH-1L. a** Two orthogonal views of the peripheral arm of NDH-1L in complex with Fd. Proteins are colored as in Fig. 1b and ion-sulfur clusters are shown as spheres. **b** Electrostatic surface potential of the Fd-binding pocket at the peripheral arm of the NDH-1L complex (positive potential, blue; negative potential, red). Fd is presented in cartoon model and colored in teal. **c** Close-up views of key Fd-interacting residues of NdhI, NdhH, NdhV, and NdhO. **d** The Fd-binding pocket of NDH-1L highly resembles that of PSI (left). Fd binds to NdhI in a similar manner as Fd to PsaC (middle). The distance and angular relationships between the [2Fe-2S] cluster in Fd and [4Fe-4S] clusters in NDH-1L and PSI (right). Arrows indicate the direction of electron flow. **e** The overall electrostatic surface potential of the Fd-binding pockets is highly positive in NDH-1L and close to neutral in PSI.

NDH-CET[18,34,35]. To further investigate the roles of OPS regulatory subunits in short-term response to high-light stress, we individually deleted each regulatory subunit in the model cyanobacterium *Synechocystis* sp. strain PCC 6803 (hereafter referred to as *Synechocystis* 6803) and monitored the NDH-CET activity by the chlorophyll fluorescence analysis[7]. Deletion of NdhL almost completely abolished the NDH-CET activity to the same extent as deletion of the essential core subunit NdhI (Fig. 4a). This result was in accordance with our hypothesis that NdhL plays an essential role in stabilizing the PQ-binding cavity (Fig. 2d, e).

Compared with NdhL, we found that the NDH-CET activity was considerably but not completely suppressed in NdhV-defective cells, suggesting that NdhV is required for the short-term response to high-light stress (Fig. 4a). To test this idea, we monitored the in vitro Fd-dependent PQ reduction activity in Δ*ndhV* thylakoid membranes[7,36] and found that its NDH-CET activity was much lower than that in the wild-type (WT) strain (Fig. 4b). Addition of purified NdhV immediately increased the activity in an NdhV dosage-dependent manner (Fig. 4b), suggesting that NdhV is a positive regulator of NDH-CET. To further examine the function of NdhV, we used the M55 (the Δ*ndhB* strain of *Synechocystis* 6803) mutant thylakoid membranes that exhibit a complete loss of the Fd-dependent PQ reduction activity due to the lack of functional NDH-1 complexes (Fig. 4c)[6,37]. This activity can be partially restored when supplemented with purified *T. elongatus* NDH-1LΔV and can be further alleviated by addition of purified *T. elongatus* NdhV

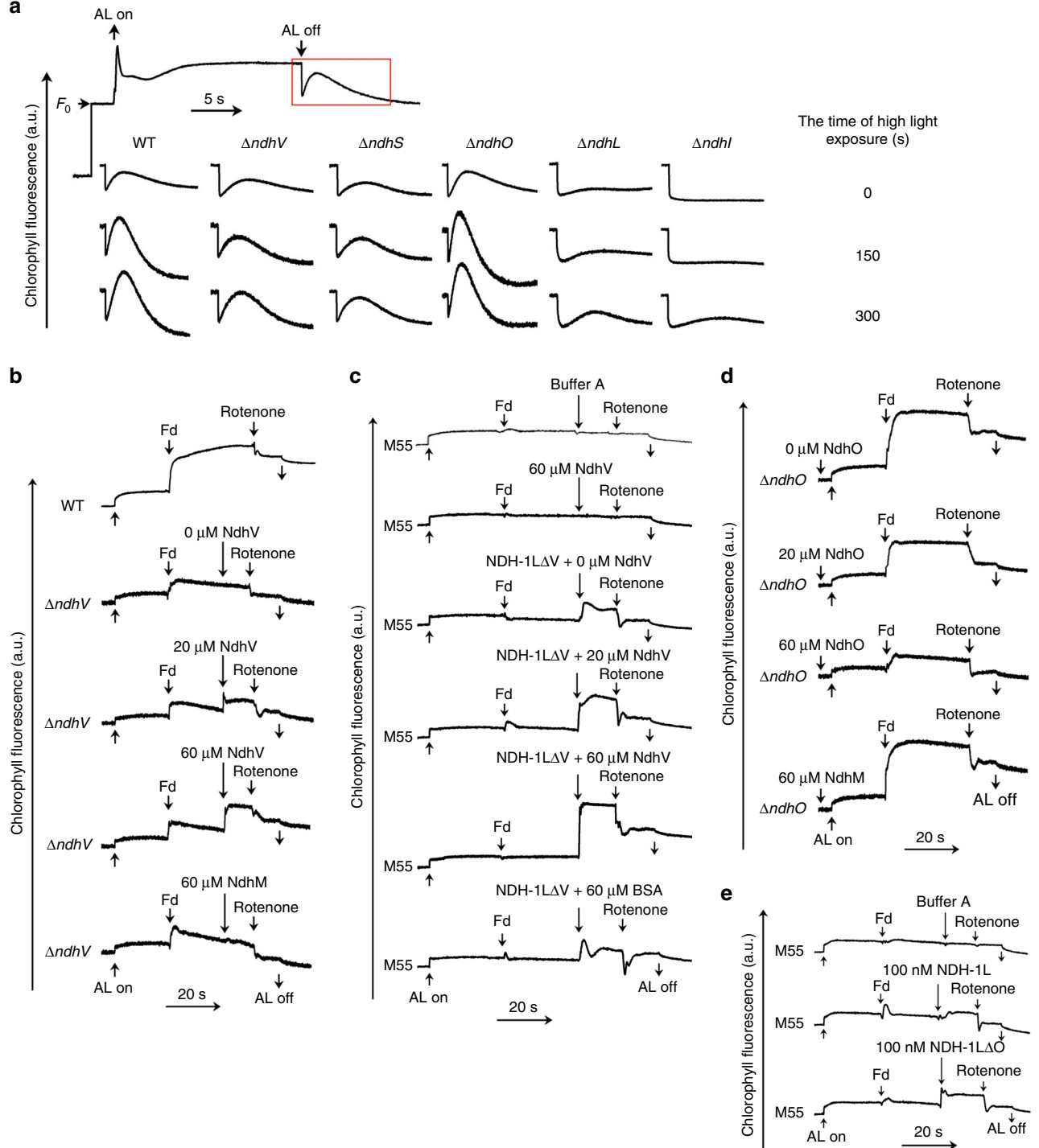

**Fig. 4 Regulatory functions of four OPS regulatory subunits. a** The top curve shows a typical kinetics of chlorophyll fluorescence in WT *Synechocystis* 6803. After WT and five defective mutants of *Synechocystis* 6803 (Δ*ndhV*, Δ*ndhS*, Δ*ndhO*, Δ*ndhL*, and Δ*ndhI*) were exposed to high light (200 μmol photons m$^{-2}$ s$^{-1}$), their short-term response was monitored by the subsequent change in the chlorophyll fluorescence level indicated in the red box region. Actinic light (AL; 620 nm; 45 μmol photons m$^{-2}$ s$^{-1}$) was switched on and off at the indicated time. **b**, **c** In vitro analysis of NdhV in Fd-dependent plastoquinone reduction. Thylakoid membranes were isolated from NdhV-defective mutant (Δ*ndhV*) cells (**b**) and the M55 mutant (**c**) of *Synechocystis* 6803. The plastoquinone reduction activity was monitored by the chlorophyll fluorescence under illumination with actinic light (AL; 620 nm, 918 μmol photons m$^{-2}$ s$^{-1}$) and inhibited by the specific inhibitor rotenone. Purified Fd, NdhV alone, NDH-1LΔV complex mixed with NdhV or NdhM, and rotenone were added at the indicated time. NdhM or BSA was used as a negative control. **d**, **e** In vitro analysis of NdhO in Fd-dependent plastoquinone reduction. Thylakoid membranes were isolated from NdhO-defective mutant (Δ*ndhO*) cells (**d**) and the M55 mutant (**e**) of *Synechocystis* 6803. Purified Fd, NdhO, NdhM, NDH-1L complex or NDH-1LΔO complex, and rotenone were added at the indicated time. NdhM or buffer as used as a negative control.

(Fig. 4c). Notably, in the absence of NDH-1LΔV, NdhV even at the highest concentration had no effect on the activity, confirming that the stimulation by NdhV is NDH-1L specific (Fig. 4c).

Given that NdhV is a transient subunit of NDH-1L, we propose that NdhV could function as an Fd-binding cofactor, guiding the association of Fd to the NDH-1L complex. In addition, NdhV may also accelerate the electron transfer rate by helping stabilize the Fd–NdhI interface in an electron transfer-competent conformation (Fig. 3d). To test this idea, we incubated NDH-1LΔV with Fd without NdhV and subjected the mixture for cryo-EM analysis (Supplementary Fig. 10). The 5.5-Å reconstruction showed that there was no detectable density for Fd in >85% of the particles in the micrographs (Supplementary Figs. 11 and 12a and Supplementary Table 2). In sharp contrast, EM densities for Fd and NdhV were clearly visible for ~76% of particles when NdhV was also included for EM sample preparation (Supplementary Figs. 2 and 12b). These results support the notion that NdhV likely plays a key role in assisting Fd binding to the NDH-1L complex. Further studies are needed to fully understand the in vivo function of NdhV in NDH-CET.

NdhS is a constitutive component of NDH-1L and adopts an SH3-like fold (Supplementary Fig. 13a). It has been considered as the Fd-binding subunit in NDH-1L[13,14,38]. We indeed observed that deletion of NdhS caused a similar level of defect in the NDH-CET activity as in ΔndhV cells in response to high-light irradiation (Fig. 4a). However, the ~25-Å distance between Fd and NdhS in the Fd-NDH-1L complex structure suggests that it is unlikely NdhS plays a major role in direct Fd binding, even though the C-terminal 10-residue tail of NdhS is not visible in the EM density (Supplementary Fig. 13b). It is more likely that NdhS makes its contribution indirectly as a foothold holding the transient NdhV in the NDH-1L complex for Fd binding (Fig. 3a, c). This structural information is in accordance with our previous data that NdhS is required for accumulation of NdhV in the thylakoid membranes but not vice versa[35].

In contrast to NdhV and NdhS, deletion of NdhO, which is also at the Fd-binding site, led to an obvious increase of NDH-CET in Synechocystis 6803 cells under high-light conditions (Fig. 4a), indicating that NdhO plays a different role from NdhV and NdhS. Unlike NdhV, addition of purified NdhO to the ΔndhO thylakoid membranes suppressed the Fd-dependent PQ reduction activity (Fig. 4d). We further purified NDH-1 complexes from the WT (NDH-1L) and ΔndhO (NDH-1LΔO) Synechocystis 6803 strains and supplemented them to the M55 mutant thylakoid membranes (Fig. 4e). We found that the activity of M55 was partially restored when the NDH-1L complex was added and it can be further alleviated when supplemented with the NDH-1LΔO complex (Fig. 4e). Taken together, these results imply that NdhO might be a negative regulator of NDH-CET activity. Future more quantitative studies are required to confirm this hypothesis and fully understand the regulatory function of NdhO in NDH-CET.

**Regulatory mechanism of NDH-CET during high-light adaptation.** To further investigate the effect of long-term high-light exposure on NDH-CET, we monitored the gene expression pattern of all NDH-1 subunits. Consistent with previous studies[15,34], reverse transcription–quantitative real-time polymerase chain reaction (RT-qPCR) analysis clearly revealed that, except for NdhO, the transcriptional levels of other subunits of NDH-1M and of all the components of the $CO_2$-uptake variable module NDH-1S were evidently induced by long-term high-light irradiation (Fig. 5a, b and Supplementary Table 3). This mRNA expression pattern was also consistent with the changes in protein

levels revealed by western blot analysis (Fig. 5c), implying that the NDH-1MS complex (without NdhO) was induced by the long-term high-light exposure. It is likely that OPS subunits NdhL, NdhS, and NdhV play the same regulatory roles in the NDH-1MS complex as in NDH-1L to accelerate the NDH-CET activity. To further confirm this observation, we performed blue native (BN)-PAGE and western blot analyses. An immunoblotting band that corresponds to the NDH-1MS complex was clearly induced by high-light exposure, although the majority of this complex had disassociated into the NDH-1M (common module) and NDH-1S (variable module) subcomplexes (Fig. 5d). Notably, NdhO was hardly detected in NDH-1MS and its subcomplex NDH-1M (Fig. 5d), further implying that NdhO likely is not a component of the NDH-1MS complex induced by long-term high-light irradiation. Compared with NDH-1MS, the NDH-1L complex was constitutively and stably expressed under conditions of long-term high-light irradiation, as revealed by the level of NdhD1, a NDH-1L complex-specific component (Figs. 1a and 5a). Taken together, we propose that the NDH-1MS complex without NdhO is induced to accelerate the NDH-CET activity by long-term high-light irradiation.

## Discussion

Photosynthetic NDH-1L and respiratory complex I share a conserved L-shaped skeleton[39] and were suggested to originate commonly from a group 4 membrane-bound [NiFe] hydrogenase that accepts electron from Fd[40]. During evolution, however, respiratory complex I and photosynthetic NDH-1L developed different catalytic reactions[8]. An NADH-binding module consisting of three subunits is capable of oxidizing NADH in the respiratory complex I[19,23,41–43]. But the counterpart of this NADH-binding-module is absent in the photosynthetic NDH-1L complex. Consistent with previous predictions, our structural data confirm that the photosynthetic NDH-1L complex retains an original electron input module that accepts electrons from Fd (Fig. 1b and Supplementary Fig. 6).

In respiratory complex I, NADH donates two electrons, via flavin mononucleotide (FMN) and the chain of Fe-S clusters, to the quinone molecule[19,23,41–43]. The transfer of these two electrons is coupled to the translocation of four protons across the membrane[23,41–43]. Based on the structural information, it was proposed that the electron transfer occurs in the sequence of NADH-FMN-N3-N1b-N4-N5-N6a-N6b-N2-quinone, in which redox centers N3-N1b-N4-N5, N6a-N6b, and N2 correspond to Fe-S clusters coordinated by the Nqo1-Nqo3, Nqo9, and Nqo6 subunits, respectively[19]. The NDH-1L structure clearly shows that subunits NdhI and NdhK, respectively, correspond to Nqo9 and Nqo6 in respiratory complex I[14] and that N6a coordinated in NdhI accepts electrons from photoreduced Fd (Fig. 3d and Supplementary Fig. 6). In contrast, counterparts of subunits Nqo1–Nqo3 in respiratory complex I are absent in cyanobacterial NDH-1L. As a consequence, the photosynthetic NDH-1L complex had developed a shorter electron transfer chain Fd-N6a-N6b-N2-PQ, which was inherited from its [NiFe] hydrogenase ancestor[8].

Four types of NDH-1, NDH-1L, NDH-1L', NDH-1MS, and NDH-1MS', have been proposed in cyanobacteria (Fig. 1a)[8–10]. Under normal growth conditions, the major type of NDH-1 is NDH-1L[37], which has a conserved L-shaped skeleton similar to that of respiratory complex I (Fig. 1b)[13,14]. When cyanobacterial cells are transferred to environments with low level of $CO_2$, the L-shaped NDH-1L complex is still constitutively expressed but a U-shaped NDH-1MS complex is significantly induced (Fig. 5d)[9,37,44]. Consequently, the NDH-1MS complex becomes the major type of NDH-1 to reinforce its $CO_2$ acquisition

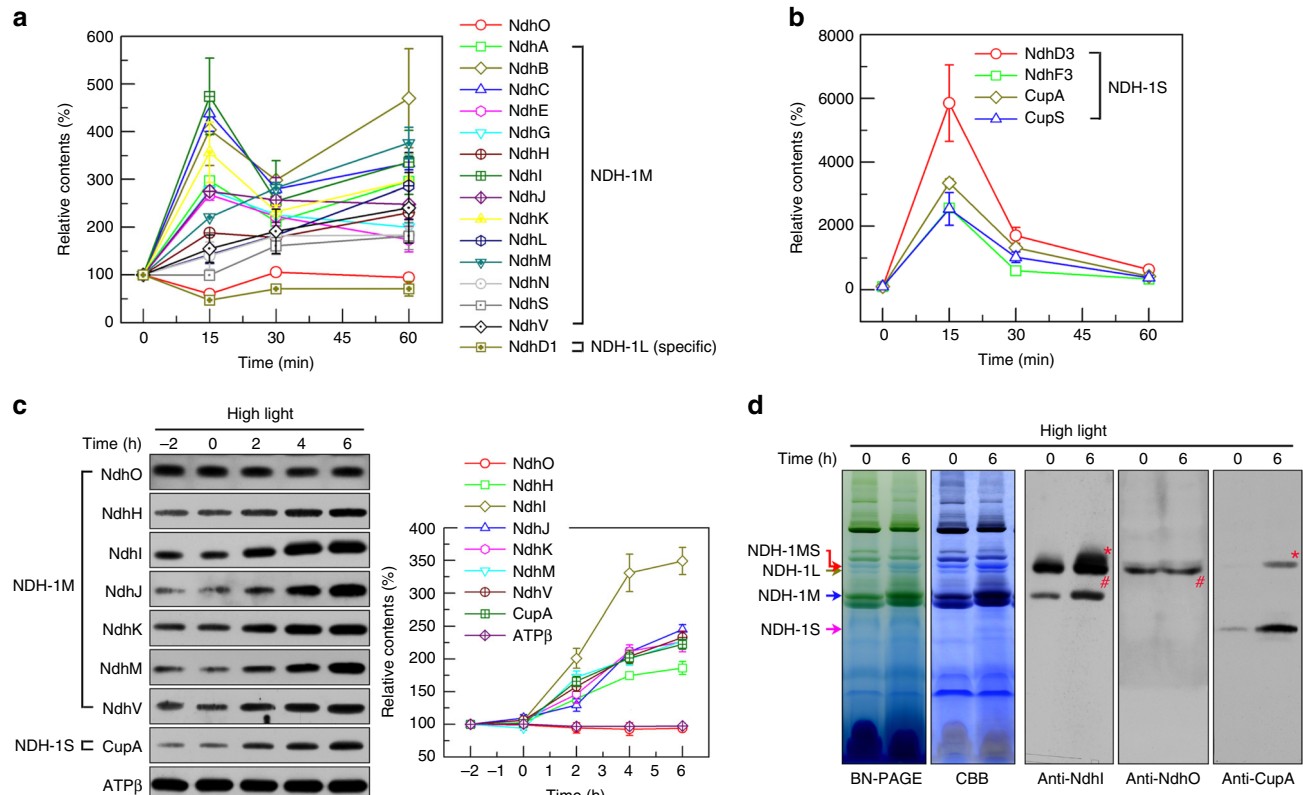

**Fig. 5 Regulatory mechanism of NDH-1 in response to high-light exposure. a, b** Time course of transcription profiles of components of NDH-1M (**a**) and NDH-1S (**b**) after cells were transferred from growth-light to high-light conditions. **c** Time course of protein-level profiles of components of NDH-1M and NDH-1S. ATPβ was used as a control. Quantification of the protein components of NDH-1M and NDH-1S is shown on the right. **d** Detection of the NDH-1MS complex and its subcomplexes NDH-1M and NDH-1S before and after 6-h high-light exposure. Red, green, blue, and pink arrows indicate the positions of NDH-1MS (denoted by asterisk (\*)), NDH-1L (denoted by hash (#)), NDH-1M, and NDH-1S, respectively. Error bars in the graph represent standard errors of the mean (SEM).

function required for the Calvin–Benson cycle[37]. The NDH-1MS complex can also be significantly induced by high-light irradiation and has been suggested to be mainly involved in PSI CET, necessary for balancing the ATP/NADPH ratio required for the Calvin–Benson cycle[11,37]. Based on these observations, we propose that under environmental stresses the NDH-1MS complex is capable of optimizing photosynthesis and decreasing ROS production via its two functions, $CO_2$ acquisition and NDH-CET.

In contrast to NDH-1L and NDH-1MS, much less is known about the other two NDH-1 complexes. The NDH-1MS' complex has been reported as a constitutive $CO_2$-uptake system expressed under various environmental stresses[45,46]. Notably, DNA microarray analysis showed that the specific component NdhD2 of the NDH-1L' complex is significantly induced by high-light irradiation[15], suggesting that NDH-1L' might be an inducible complex similar to the NDH-1MS complex. However, NDH-1MS' and NDH-1L' complexes have not been biochemically isolated yet, possibly because of their low abundance and/or fragile character. Further studies will be needed to unravel the importance of the NDH-1L' and NDH-1MS' complexes under environmental stresses.

Our structural and functional data reported here provide an integrated antioxidant defense mechanism of cyanobacteria against excess light in their natural habitat. In this model, the transient regulatory subunit NdhV plays an important role in this process. We propose that, under growth-light conditions, the low level of photoreduced Fd from PSI only needs a fraction of NdhV to assist its binding to the constitutively expressed NDH-1L, restraining the electron transfer to balance the ATP/NADPH ratio required by Calvin–Benson cycle (Fig. 6). When cyanobacteria are challenged by a short-term high-light exposure, we propose that, although the expression level of NdhV remains largely unchanged (Fig. 5a), the available NdhV molecules are sufficient to associate with increased photoreduced Fd to bind to the NDH-1L complex and to accelerate the electron transfer rate for the need of efficient photosynthesis (Fig. 6). If the high-light exposure extends to become a long-term stress, cyanobacteria switch on the expression of NdhD3, NdhF3, CupA, and CupS and the assembly of the NDH-1MS complex to further accelerate the electron transfer rate (Fig. 6). We propose that this exquisite, hierarchical operation of cyanobacteria in response to different levels of high-light irradiation is a key regulatory mechanism for accelerating PSI CET and reducing ROS production necessary for the survival of cyanobacteria in aerobic environment and ultimately for opening up the evolution of complex organisms on Earth.

## Methods

**Cell culture.** WT strain of *T. elongatus* BP-1 (NIES-2133) was grown at 45 °C in BG-11 medium buffered with 5 mM Tris–HCl (pH 8.0) and bubbled with 2% (v/v) $CO_2$ in air. Continuous illumination was provided by fluorescence lamps at 50 μmol photons $m^{-2} s^{-1}$.

Glucose-tolerant strain of WT *Synechocystis* sp. PCC 6803 (ATCC 27184) and its mutants, ΔndhI[47], M9 (ΔndhL)[48], M55 (ΔndhB), ΔndhO, ΔndhS, and ΔndhV, were cultured at 30 °C in BG-11 medium buffered with 5 mM Tris–HCl (pH 8.0) and bubbled with 2% (v/v) $CO_2$ in air. The mutant strains were grown in the presence of appropriate antibiotics under continuous illumination by fluorescence lamps at 40 or 200 μmol photons $m^{-2} s^{-1}$ for high-light exposure for short-term (150 or 300 s, respectively) and long-term (2, 4, or 6 h, respectively)

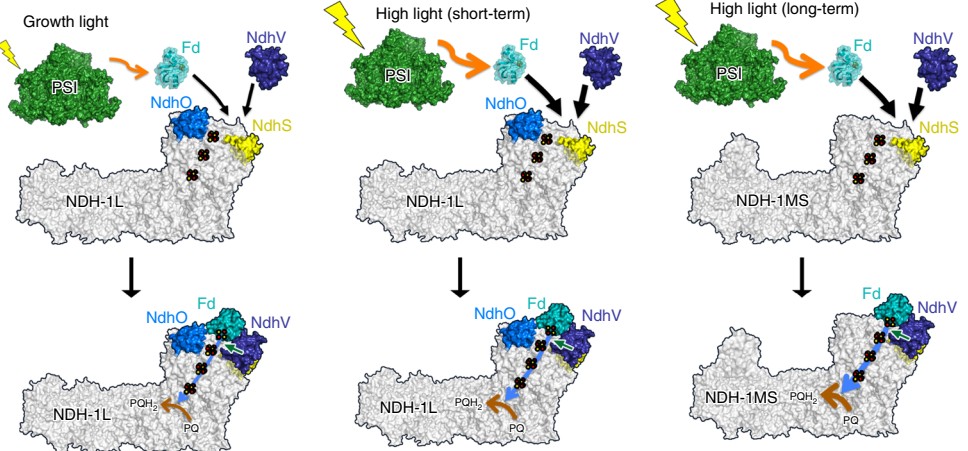

**Fig. 6 A schematic model for NDH-1-mediated PSI CET in response to different levels of light exposure.** The color scheme is as follows: NdhV, deep blue; NdhS, yellow; NdhO, marine; the rest of NDH-1L and NDH-1MS, gray; Fd, teal; and PSI, green. The electron transfer pathway [2Fe-2S]-N6a-N6b-N2-PQ and the reduction of PQ to PQH$_2$ are denoted by sky-blue and orange arrows, respectively. The high-light intensity increases the level of photoreduced Fd from PSI (orange arrow between PSI to Fd) and the level of Fd bound to the NDH-1 complex (black arrow between Fd and NDH-1L or NDH-1MS). Positive regulation of NDH-CET by NdhV is indicated by a green arrow.

**Chlorophyll fluorescence.** Cells were harvested at the logarithmic phase (OD$_{730}$ = 0.6-0.8) with centrifugation and the cell pellets were suspended in fresh BG-11 medium at a chlorophyll $a$ concentration of 10 μg mL$^{-1}$. Subsequently, cells were exposed to actinic light (AL; 620 nm; 45 μmol photons m$^{-2}$ s$^{-1}$) for 30 s. After AL was turned off, the transient increase in the chlorophyll fluorescence level was monitored using a Dual-PAM-100 system (Walz, Effeltrich, Germany) as an indication of NDH-CET activity[49].

**Isolation and solubilization of crude thylakoid membranes.** Cells were harvested at the logarithmic phase (OD$_{730}$ = 0.6–0.8) with centrifugation and the cell pellets were washed twice with fresh BG-11 medium, and the thylakoid membranes were isolated as described with some modifications as follows[50]. The cells were re-suspended in 50 mL of disruption buffer (20 mM BisTris–HCl, pH 6.0, 10 mM MgCl$_2$, 10 mM CaCl$_2$, 20% (v/v) glycerol) and were broken at 4 °C by a high-pressure homogenizer (PhD Technology LLC, USA) for 3 cycles at a pressure of 5000 psi. The crude extract was centrifuged at 5000 × $g$ for 5 min at 4 °C to remove the unbroken cells. By further centrifugation at 150,000 × $g$ for 30 min at 4 °C, crude thylakoid membranes were obtained from the precipitation. The crude thylakoid membranes were suspended in solubilization buffer (20 mM BisTris–HCl, pH 6.0, 10 mM MgCl$_2$, 20% (v/v) glycerol) at a final chlorophyll $a$ concentration of 1 mg mL$^{-1}$ and then were solubilized by adding 10% (w/v) $n$-dodecyl-β-D-maltoside (DDM) dropwise to 1% with constant stirring on ice for 1 h. Insoluble sample was removed by centrifugation at 20,000 × $g$ for 15 min at 4 °C.

**Purification of the NDH-1LΔV complex from $T.$ $elongatus$.** The NDH-1LΔV complex was purified from the thermophilic cyanobacterium $T.$ $elongatus$ BP-1 using a native polyhistidine tag on subunit NdhF1. It was purified at 4 °C using Ni$^{2+}$ affinity chromatography (NAC) and size-exclusion chromatography (SEC) as described with some modifications[50]. Solubilized thylakoid membranes, filtered through a 0.45-μm membrane, were applied to the NAC column equilibrated with buffer A (20 mM BisTris–HCl, pH 6.0, 100 mM NaCl, 1% (v/v) glycerol, 0.1% digitonin) and washed with 3-time column volume of buffer A. The protein samples of the NDH-1LΔV complex were eluted with 150 mM imidazole in buffer A with 2-time column volume and concentrated using the Amicon YM-100 (Millipore) system. SEC with the Superose6 Increase 10/300GL column (GE Healthcare) equilibrated with buffer A was then used for further purification and the peak fraction containing NDH-1LΔV was collected and concentrated to ~3.5 mg mL$^{-1}$.

**Purification of the NDH-1 complexes from $Synechocystis$ 6803.** The NDH-1L and NDH-1LΔO complexes were purified from the WT $Synechocystis$ sp. PCC 6803 (ATCC 27184) and its mutant Δ$ndhO$, respectively, by using a Strep tag II on subunit NdhL. They were purified at 4 °C using Strep-Tactin affinity chromatography and SEC as described with some modifications[50]. Solubilized thylakoid membranes, filtered through a 0.45-μm membrane, were applied to the Strep-Tactin resins equilibrated with buffer A (20 mM BisTris–HCl, pH 6.0, 100 mM NaCl, 1% (v/v) glycerol, 0.1% digitonin) and washed with 3-time column volume of buffer A. The protein samples of the NDH-1LΔO complex and NDH-1L (WT) were eluted with 50 mM biotin in buffer A with 2-time column volume and concentrated using the Amicon YM-100 (Millipore) system. SEC with the Super-ose6 Increase 10/300GL column (GE Healthcare) equilibrated with buffer A was

then used for further purification and the peak fraction containing NDH-1LΔO and NDH-1L (WT) were collected and concentrated to ~3.5 mg mL$^{-1}$.

**Purification of Fd and NdhV and NdhO from $Escherichia$ $coli$.** Full-length genes of Fd, NdhV, or NdhO from $T.$ $elongatus$ BP-1 (NIES-2133) and $Synechocystis$ 6803 (ATCC 27184) were cloned into the pMAL vector, which has an N-terminal fused MBP tag and transformed into the $E.$ $coli$ Rosetta (DE3) cells (TransGen Biotech Co., Ltd). The cells were cultured in Luria broth at 37 °C until the OD$_{600}$ reached about 1.0 and were then induced with 0.1 mM isopropyl-beta-D-thiogalactopyr-anoside and allowed to grow at 18 °C overnight. The cells were harvested by centrifugation and the pellets were resuspended in lysis buffer (25 mM Tris, pH 7.5, 150 mM NaCl, 10% glycerol). The cells were then lysed by sonication and the cell debris was removed by ultracentrifugation. The supernatant was mixed with amylose agarose beads (New England Biolabs) and rocked for 4 h at 4 °C before elution with 20 mM maltose. The PreScission protease was added to remove the MBP tag. The proteins were then further purified by Mono-Q and by gel filtration chromatography equilibrated with 25 mM Tris, pH 6.5, and 100 mM NaCl. The purified proteins were concentrated to 30 mg mL$^{-1}$ and stored at −80 °C.

**Electrophoresis and immunoblotting.** BN-PAGE of the purified samples were performed as described previously[51] with modification[52,53]. The purified samples were mixed with 1/10 volume of sample buffer (5% Serva Blue G, 100 mM Bis-Tris, pH 7.0, 30% (w/v) Sucrose, 500 mM ε-amino-$n$-caproic acid, and 10 mM EDTA) and then were applied to a 0.75-mm-thick, 5–12.5% acrylamide gradient gel (Hoefer Mighty Small mini-vertical unit; San Francisco, CA). The purified samples were loaded on an equal protein basis of 7 μg per lane. Electrophoresis was performed at 4 °C by increasing the voltage gradually from 50 up to 200 V during the 5.5-h run.

CN-PAGE was performed with 0.01% DDM and 0.025% deoxycholate additives to the cathode buffer as described previously[54,55]. The purified sample (10 μg of protein) was applied to a 0.75-mm-thick, 5–12.5% acrylamide gradient gel. Electrophoresis was performed at 4 °C by increasing the voltage gradually from 50 up to 200 V during the 5.5-h run. Sodium dodecyl sulfate-PAGE of crude thylakoid membranes of WT $Synechocystis$ 6803 was performed on a 12% polyacrylamide gel with 6 M urea as described previously[56].

For immunoblotting, the proteins were electrotransferred to a polyvinylidene difluoride membrane (Immobilon-P; Millipore, Bedford, MA) and detected by protein-specific antibodies using an ECL assay kit (Amersham Pharmacia, NJ) according to the manufacturer's protocol. Antibodies against NdhH (1:1000), NdhI (1:2000), NdhJ (1:1000), NdhK (1:1000), NdhM (1:1000), NdhO (1:1000), NdhV (1:1000), and ATPβ (1:2000) were raised in our laboratory. Antibody against CupA (1:1000) was provided from Professor Hualing Mi (Institute of Plant Physiology and Ecology, Chinese Academy of Sciences).

**Mass spectrometric analysis of the purified NDH-1LΔV complex.** Peptide preparation and liquid chromatography-electrospray ionization tandem mass spectrometry (LC-ESI MS/MS) analyses were performed as described[55]. A yellow NDH-1LΔV band excised from the CN gel was treated twice with 50 mM ammonium bicarbonate in 30% (v/v) acetonitrile for 10 min and 100% (v/v) acetonitrile for 15 min and then dried in a vacuum concentrator. The dried gel pieces were treated with 0.01 mg mL$^{-1}$ trypsin (sequence grade; Promega)/50 mM

ammonium bicarbonate and incubated at 37 °C for 20 h. The digested peptides in the gel pieces were recovered twice with 20 μL 5% (v/v) formic acid/50% (v/v) acetonitrile. The extracted peptides were combined and then dried in a vacuum concentrator.

LC-MS/MS analyses were performed on a Q-Exactive mass spectrometer (Thermo Scientific, Bremen, Germany) coupled with an Easy-nLC1000 HPLC system (Thermo Scientific). Trypsin-digested peptides were dissolved in 12 μL of 2% formic acid, loaded onto a C18-reversed-phase column (75 μm i.d., 15 cm) in buffer B (2% acetonitrile and 0.1% formic acid), and separated using a 90-min linear gradient from 3% to 40% of buffer C (80% acetonitrile and 0.1% formic acid) at a flow rate of 250 nL min$^{-1}$. MS data were acquired in a data-dependent mode, each full scan ($m/z$ 300–1800, resolution of 70,000 at $m/z$ 200) was followed with 10 high-energy collisional dissociation MS/MS scans ($m/z$ 150–2000, resolution of 17,500 at $m/z$ 200) for the most intense precursor ions. The raw data were processed with the Protein Discoverer software, version 1.1 (Thermo Scientific) using default parameters to generate concatenated Mascot generic files. Database searches were performed using an in-house MASCOT server (version 2.2) against a UniProt database of *T. elongatus* BP-1 proteins. The search criteria allowed for one missed cleavage of trypsin, oxidation of methionine, and 6-ppm and 0.1-Da mass accuracies for MS and MS/MS modes, respectively. The decoy database searches were also performed in parallel, and peptides <1% false discovery rate were accepted.

**Cryo-EM data acquisition and image processing.** For NDH-1LΔV, 3 μL of sample (~3.2 mg mL$^{-1}$) was applied to glow-discharged Quantifoil R 1.2/1.3 copper grids and blotted for 2–3 s at 100% humidity at 10 °C in the chamber of FEI Vitrobot IV. For the Fd-NDH-1L complex/Fd(-)-NDH-1L, the purified Fd and NdhV or purified Fd alone were mixed with NDH-1LΔV (10 μM) with a molar ratio of ~20:1, incubated for 20 min on ice, and then applied on glow-discharged Quantifoil R 1.2/1.3 copper grids. Immediately afterwards, sample was snap frozen in liquid ethane. Grids were transferred to an FEI Titan Krios EM operated at 300 KV. For the NDH-1LΔV and Fd-NDH-1L complexes, images were recorded by a direct electron detector FEI Falcon-III at calibrated magnification of 75,000 in the electron counting mode with a pixel size of 1.09 Å. EPU software (FEI) was used for fully automated data collection. Each stack was exposed for 53 s with dose of 0.95 $e^-$ pixel$^{-1}$ s$^{-1}$, resulting in a total of 32 frames per stack. The total dose rate was ~40 $e^-$ Å$^{-2}$ for each stack. The defocus range was set from −1.8 to −2.7 μm. For NDH-1LΔV mixed with Fd (hereafter referred to as (Fd)-NDH-1LΔV), images were recorded by K3 detector at calibrated magnification of 81,000 with a pixel size of 1.09 Å. Serial EM was used for fully automated data collection. Each stack was exposed for 3.2 s with an exposing time of 0.1 s per frame, resulting in a total of 32 frames per stack. The total dose rate was ~50 $e^-$ Å$^{-2}$ for each stack. The defocus range was set from −1.5 to −3.0 μm.

For NDH-1LΔV images, a total of 7157 micrographs were collected in three datasets, including 1472, 2755, and 2930 micrographs, respectively. In all, 2–32 frames in each image were aligned and summed by using the whole-image motion correction program MotionCor2[57]. Dose weighting process was performed on micrographs with MotionCor2. The contrast transfer function (CTF) parameters were estimated by Gctf[58] and all the three-dimensional (3D) reconstructions were performed with RELION 3[59–61]. We manually picked about 1000 particles to generate the templates for particle auto-picking using Gautomatch (http://www.mrc-lmb.cam.ac.uk/kzhang/Gautomatch/) for further processing. Low-quality images and false-positive particles were successfully removed by the reference-free two-dimensional (2D) classification, yielding 163,085, 236,636 and 294,044 particles for each dataset. Then the first round 3D classification was performed for each dataset individually using a 50 Å low-pass filtered initial model generated from a previous model of *T. thermophilus* complex I (PDB 4HEA). The particles of good quality (101,682, 121,328 and 216,936 particles for each dataset, respectively) were then selected and combined as a new dataset of 439,946 particles to be subjected to the final refinement. The initial 3D auto-refine procedure was performed, resulting in an overall structure at a resolution of 4.1 Å. The second round of refinement was performed by applying a soft mask being 4.1 Å and improved resolution to 4.0 Å. The overall resolution was finally improved to 3.6 Å by performing the final 3D auto-refine after CTF refinement procedure in RELION 3. All resolutions are based on the gold-standard (two halves of data refined independently) Fourier shell correlation (FSC) = 0.143 criterion substitution[62]. RELION 3 was used to estimate the local resolution variations of the density map.

For Fd-NDH-1L images, a total of 5435 micrographs were collected in two datasets, including 2304 and 3131 micrographs, respectively. Two-to-32 frames in each image were aligned and summed by using the whole-image motion correction program MotionCor2. Dose weighting process was performed on micrographs with MotionCor2. The CTF parameters were estimated by Gctf and all the 3D reconstructions were performed with RELION 3. The particle picking template and strategy is the same as NDH-1LΔV images, yielding 263,556 and 322,181 particles. One round reference-free 2D classification and one round 3D classification were preformed; 338,822 particles were selected to high-resolution refinement. Two round 3D refinement were performed, resulting in an overall structure at a resolution of 4.1 Å. The overall resolution was finally improved to 3.2 Å by performing the final 3D auto-refine after CTF refinement and Bayesian polishing procedure in RELION 3. Furthermore, in order to improve the density of Fd and

NdhV, a local approach was employed. Except for Fd and NdhV, projections were subtracted from the rest region of the complex in experimental particle images[63]. Because the region of Fd and NdhV is too small for masked refinement, 3D classification was performed with a mask around the region of Fd and NdhV, without image alignment on the subtracted experimental images. After 3D classification, the major class of 257,166 particles was selected and reverted to original particle images for further processing. After CTF refinement and Bayesian polishing, final 3D refinement resulted in an overall structure at a resolution of 3.4 Å, with improved density of Fd and NdhV. All resolutions are based on the gold-standard (two halves of data refined independently) FSC = 0.143 criterion substitution. Local resolution estimations were calculated using RELION 3.

For (Fd)-NDH-1LΔV images, a total of 2552 micrographs were collected. Two-to-32 frames in each image were aligned and summed by using the whole-image motion correction program MotionCor2. Dose weighting process was performed on micrographs with MotionCor2. The CTF parameters were estimated by Gctf and all the 3D reconstructions were performed with RELION 3. The particle picking template and strategy is the same as NDH-1LΔV images, yielding 389,173 particles. Particle sorting and reference-free 2D classification and 3D classification were performed to remove contaminants and noisy particles. The most populated 3D class (109,984 particles) was subsequently selected for the final 3D auto-refinement with a soft mask including the protein and detergent regions. This generated a map with an overall resolution of 6.54 Å. The overall resolution was finally improved to 5.2 Å by performing the final 3D auto-refine after CTF refinement and Bayesian polishing procedure in RELION 3. The density of Fd is absent in 5.2-Å near-atomic resolution map. To further confirm whether Fd was bound to NDH-1LΔV complex or not, exactly the same local approach was employed. Briefly, after 3D classification, without image alignment, the most populated class (85.4%) of 93,926 particles was selected for further processing. After CTF refinement and Bayesian polishing, final 3D refinement resulted in an overall structure at a resolution of 5.5 Å lack of the density of Fd. All resolutions are based on the gold-standard (two halves of data refined independently) FSC = 0.143 criterion substitution. Local resolution estimations were calculated using RELION 3.

**Model building and refinement.** De novo atomic model building and rigid docking of homologous structure modeling are combined to generate an atomic model for the entire NDH-1L from *T. elongatus* BP-1. For the 11 conserved core subunits, the initial homology models were generated manually based on the *T. thermophilus* structure with side chains rebuilt to *T. elongatus* BP-1 sequence using the COOT software[64]. Information of multiple sequence alignments and secondary structure predictions for all subunits were very helpful during this part of model building.

De novo model building was performed for NdhL, NdhP, and NdhQ as well as NdhM, NdhN, NdhO, NdhS, and NdhV, which lack a structural model. It is predicted that NdhP, NdhQ, and NdhL contain 1, 1, and 2 transmembrane helices, respectively. This led us to first build the characterized helices in NdhP, NdhQ, and NdhL, and then other elements were assigned. For the hydrophilic subunits, NdhS was first built by docking its *Synechocystis* sp. strain PCC 6803 homolog structure (PDB 2JZ2) into the electron map. Information of the predicted secondary structure of NdhM, NdhN, NdhO, and NdhV and chemical features of their amino acids were taken into account and the building process was greatly aided by the EM density of bulky residues, such as Phe, Tyr, Trp, and Arg. All the characterized helices and β-sheet in NdhM, NdhN, NdhO, and NdhV were first built and then other elements were assigned. The initial models were adjusted to cryo-EM density or built manually in Coot. β-Carotene (BCR) and lipids were tentatively assigned on the basis of appearance in the density as SQD, LHG, and DGD, known to co-purify with other photosynthetic complex in *T. elongatus* BP-1 (PDB 5KAF and PDB 5ZF0).

Multiple sequence alignments for all subunits were performed with the program T-coffee server (http://tcoffee.crg.cat/apps/tcoffee/do:mcoffee)[65]. The position of transmembrane helices in the analyzed sequences was predicted using the TMHMM server v2.0 (http://www.cbs.dtu.dk/services/TMHMM/)[66]. Protein secondary structures are predicted by PSIPRED[67], model building was performed in Coot, and final model refinement was carried out using phenix. real_space_refine[68] with secondary structure and geometry restrains to prevent over-fitting. The structure of the cyanobacteria NDH-1L complex was validated by using MolProbity[69]. The Quinone-binding cavity was predicted using CAVER 3.0[70] with a 1.4-Å probe radius, starting from the side chain O atom of Nqo4-Tyr87 (*T. thermophilus*), and NdhH- Tyr72 (*T. elongatus*). Images of the EM density map and the structural model were prepared by using Chimera[71] and PyMOL[72], respectively.

**Absorption spectrum of the purified NDH-1LΔV complex.** The purified NDH-1LΔV complex was separated by CN-PAGE. A yellow NDH-1L band was excised from the CN gel and then was analyzed by a spectrophotometer (UV3000; Shimadzu) through positioning it into a cuvette. The full wavelength scan of β-carotene (10 μM; Sigma) is conducted as a control of its typical characteristic peaks.

**Mass spectrometric analysis of lipids**. The lipids binding to NDH-1LΔV complex were extracted by adding 300 μL of chloroform/methanol (2:1; v/v) into 30 μL of NDH-1LΔV complex solution. The mixture was placed in an ultrasonic cleaner for 5 min and centrifuged at $3000 \times g$ for 3 min. The organic phase (yellow, lower layer) was pipetted into a new tube and was washed with brine twice. The organic solvent was evaporated using a vacuum concentrator, and the lipid extracts were re-dissolved in 30 μL of chloroform/methanol (1:2; v/v) for mass spectrometric analysis. The lipid extracts were directly infused into a high-resolution mass spectrometer Q-Exactive HF (Thermo Scientific, Bremen, Germany) equipped with a nano-electrospray source at a flow rate of 300 nL min$^{-1}$. The full mass spectrometry scan ($m/z$ 300–2000) was acquired in negative mode with a resolution of 240,000.

**RT-qPCR analysis**. The RNA used for RT-qPCR analysis was derived from the high-light-induced cells of *Synechocystis* 6803. Three replicates of independently high-light-induced materials were used. The RNA was extracted using TRIzol (Invitrogen), purified using an RNeasy Mini Kit (Qiagen), and used as a template for cDNA synthesis. One microgram RNA was used as template for the RT of 20 μL cDNA using the PrimeScript™ RT reagent Kit with gDNA Eraser (Perfect Real Time) (TAKARA). Two microliters of the RT reaction were used to analyze the gene expression level by RT-qPCR and normalized to that of 16S rRNA. The primer sequences are presented in Supplementary Table 3. The TB Green™ *Premix Ex Taq*™ II (Tli RNaseH Plus) (TAKARA) reagent was used for the real-time PCR, and the reaction was performed in a 20-μL volume. The real-time PCR was performed in the LightCycler 480 (Roche), and each experiment was repeated twice, each in triplicates.

**In vitro assay of Fd-dependent PQ reduction**. In vitro assay of Fd-dependent PQ reduction was performed as described previously[73] with some modifications. In brief, the thylakoid membranes collected from the WT *Synechocystis* 6803 were suspended in buffer D (0.5 M sorbitol, 10 mM MgCl₂, 10 mM NaCl, 10 mM HEPES, and 5 mM sodium phosphate, pH 7.5) at a final chlorophyll *a* concentration of 80 μg mL$^{-1}$. Purified Fd (20 μM) was added, and an increase in chlorophyll fluorescence was recorded using a Dual-PAM-100 system (Walz). To test whether Fd-dependent PQ reduction mainly comes from NDH-1-dependent PQ reduction, 50 μM rotenone (Sigma-Aldrich) was added to the assay to inhibit Fd-dependent PQ reduction; also, Fd-dependent PQ reduction was assayed in NdhV/NdhO/NDH-1 complex-defective mutant ΔndhV/ΔndhO/M55 of *Synechocystis* 6803, respectively. For in vitro reconstitution of Fd-dependent PQ reduction in the thylakoid membranes of ΔndhV/ΔndhO, purified NdhV/NdhO (0, 20, 60 μM) from *T. elongatus* BP-1 or from *Synechocystis* 6803 was added, and 60 μM of NdhM was used as the control. For in vitro reconstitution of Fd-dependent PQ reduction in the thylakoid membranes of M55, purified NDH-1LΔO (100 nM) or NDH-1L (100 nM) from ΔndhO or WT *Synechocystis* 6803 strains was added or purified NDH-1LΔV (150 nM) and NdhV (0, 20, 60 μM) from *T. elongatus* BP-1 was added, and 60 μM of NdhV was used as the control.

**Reporting summary**. Further information on research design is available in the Nature Research Reporting Summary linked to this article.

## Data availability

Data supporting the findings of this manuscript are available from the corresponding authors upon reasonable request. A reporting summary for this article is available as a Supplementary Information file. The source data underlying Fig. 5a–d and Supplementary Figs. 1a, 3a, b, and 8a are provided as a Source Data file. The cryo-EM 3D maps of Fd-NDH-1L, NDH-1LΔV, and (Fd)-NDH-1LΔV were deposited in EMDB database with accession codes EMD-0849, EMD-0850 and EMD-0851, respectively. The atomic models of Fd-NDH-1L and NDH-1LΔV were deposited in PDB with accession codes 6L7O and 6L7P, respectively.

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

## Acknowledgements

We thank the staff members of the Electron Microscopy System and Mass Spectrometry System at Shanghai Institute of Precision Medicine for providing technical support. We thank M. Cao and S.B. Li for their help on cryo-EM data collection and analysis. We thank Dr. M. Marcia from EMBL Grenoble for kind help on manuscript discussion and preparation. This work was supported by grants from the National Natural Science Foundation of China (31525007 to M.L., 31570235 and 31770259 to W.M., 81602388 to C.Z.), Science and Technology Commission of Shanghai Municipality (18DZ2260500 to W.M.), and Shanghai Municipal Education Commission−Gaofeng Clinical Medicine Grant Support (20181711 to J.W.).

## Author contributions

C.Z. and Z.R. purified the NDH-1L complex. C.Z. and J.S. prepared cryo-EM specimens, collected data sets, and determined the structure. J.W. carried out model building and refinement. R.L. made the mass spectrometric analysis. Z.W. analyzed the Q-PCR data. Z.R and J.Z. carried out physiological and biochemical analysis. All authors contributed to data interpretation and the writing of the manuscript. M.L., W.M., and C.Z. wrote the manuscript. M.L., W.M., C.Z., and J.W. initiated and orchestrated the project.

## Competing interests
The authors declare no competing interests.
