## [Peer Review File · Nature Communications]

Reviewers' Comments:

Reviewer #1:

Remarks to the Author:

This manuscript by Zhang et al describe the structure of NDH1 from thermophilic cyanobacteria together with its electron donor Fd. NDH1 is one of the conserved components of oxygenic photosynthesis and plays an import role in cyclic electron flow and the adaptation of photosynthetic organisms to stress conditions. Two recently published structures did not resolve Fd as part of NDH1 and in one case raised some additional controversy around the electron transport cofactors in the complex. The scientific interest in identifying the Fd binding site and establishing a consensus on the composition of the electron transport factors is great.

The manuscript reports on three structures, one of them contains heterologously expressed Fd and NdhV. The scope of work reported here, the scientific merit of the new results and the quality of work merits publication in Nature Communication.

The authors mention the two previous NDH1 structures in passing on page 3, line 60. This is not acceptable and a clear and direct reference to the previously published structures should be made. The quality of the structural work appears sufficient, however some corrections to resolution reporting methods are required and these are detailed below.

In addition, the provided validation reports are not final as they do not contain a PDBID and EMBID codes which are also missing from the manuscript. Before approval, the authors must provide PDB codes and deposition validation reports.

In addition to their structural work, the authors constructed deletion mutants of several of the photosynthetic specific subunits of NDH1. Deletion mutants in most of these subunits were already examined and most of the results presented in figure 4a are not new, this should be clearly indicated.

The authors suggest that NdhV plays a positive role in Fd binding and provide several lines of evidence to support this. The in-vitro results generally fall in line with this suggestion, but no evidence in vivo is presented.

With regards to NdhO the authors suggest that it plays a role of negative regulator of NDH1 activity. This set of results is somewhat problematic as the main evidence of the increased activity of NDH1 Δ O comes from a semi quantitative essay (Figure 4E). A more quantitative assay for NDH1 activity done on NDH1 and NDH1 purified from NdhO mutants should be done if the authors insist on including this point.

The manuscript appears to be hastingly written and in its present form and is difficult to follow. Supplementary figure 1E should contain FSC curves for phase randomized volumes as well as masked and unmasked volume and the identify of each curve should be clearly visible. The correlation between the map and model should also be reported on the same graph. Supplementary figure 3F should contain FSC curves for phase randomized volumes as well as masked and unmasked volume and the identify of each curve should be clearly visible. The correlation between the map and model should also be reported on the same graph. Supplementary figure 10D should contain FSC curves for phase randomized volumes as well as masked and unmasked volume and the identify of each curve should be clearly visible. The correlation between the map and model should also be reported on the same graph.

Minor points –

Line 67 – “Fd-N6a-N6b-N2-PQ” is used without indicating abbreviation of the iron-sulfur clusters.
Page 3, line 49 – “improves the Calvin-Benson cycle” seems like the wrong way of stating the better fit between the ATP/NADPH requirements of the CBB cycle and the output of the light reactions.

Page 4, line 97 – “Consistent with the recently reported structures of NDH-1 Δ V” should be changed to “recently reported structures of NDH-1”. As it is the authors choice to name this complex NDH-1 Δ V. Same is true for other places in the text where this is done.

Page 5, line 106 – “deactive state” should be “inactive state”

Page 7, line 153 – “Strikingly, a continuous grease belt,” should be replaced with a more

appropriate term.

Page 11, line 252 - "It has been controversial" should be corrected.

Line 342 - "Markedly, we found that the negative regulatory OPS subunit NdhO is absent in the NDH-1MS complex (Fig. 6)." - this is not shown in Fig. 6.

Reviewer #2:

Remarks to the Author:

The manuscript by Chunli Zhang et al "Structural insights into NDH-1 mediated cyclic electron transfer" describes the structure of cyanobacterial NDH complex with bound Fd. This is an important study which describes how exactly Fd interacts with NDH complex and how these interactions are regulated under different conditions by expression of specific subunits. The manuscript is generally well written and clearly illustrated, but will benefit from corrections by native English speaker. It will be of interest to the wide Nature Communications readership.

Some comments to address in revision:

Line 66 - Which four OPS regulatory subunits?

106 - Structure of ovine complex I (5LNK) is not of deactive state, but of the "open" state, likely part of the catalytic cycle along with the "closed" state (see e.g. pubmed 31492636). In the true deactive state β 1- β 2 loop is disordered.

128 - should be discussed how exactly β -carotene molecules stabilize the complex (contacts, environment).

296-299 - very convoluted sentence, should be re-phrased.

326-329 - This speculation seems unlikely, there are plenty of Fe-containing cofactors in cyanobacteria/chloroplast enzymes. The short cluster chain is simply inherited from hydrogenases.

363-368 - The difference between growth and high-light short term is not clearly stated or shown in figure - presumably NdhV expression levels are much higher in the latter case?

Point-to-point response to reviewers' comments

Reviewer #1:

This manuscript by Zhang et al describe the structure of NDH1 from thermophilic cyanobacteria together with its electron donor Fd. NDH1 is one of the conserved components of oxygenic photosynthesis and plays an import role in cyclic electron flow and the adaptation of photosynthetic organisms to stress conditions. Two recently published structures did not resolve Fd as part of NDH1 and in one case raised some additional controversy around the electron transport cofactors in the complex. The scientific interest in identifying the Fd binding site and establishing a consensus on the composition of the electron transport factors is great. The manuscript reports on three structures, one of them contains heterologously expressed Fd and NdhV. The scope of work reported here, the scientific merit of the new results and the quality of work merits publication in Nature Communication.

Thanks!

(1) The authors mention the two previous NDH1 structures in passing on page 3, line 60. This is not acceptable and a clear and direct reference to the previously published structures should be made.

Following this reviewer's suggestion, we directly cited the two recently published structures in the revised manuscript.

(Page 3, lines 60-62), "Two recently published structures of NDH-1L revealed that subunit NdhD mediates the direct interaction with the NDH-1M module and that one β -carotene molecule may be involved in stabilizing their association (Laughlin, Bayne et al. 2019, Schuller, Birrell et al. 2019)".

(Page 5, line 100), "Consistent with the recently reported structures of NDH-1 (Laughlin, Bayne et al. 2019, Schuller, Birrell et al. 2019)...".

(2) The quality of the structural work appears sufficient, however some corrections to resolution reporting methods are required and these are detailed below. In addition, the provided validation reports are not final as they do not contain a PDBID and EMDBID codes which are also missing from the manuscript. Before approval, the authors must provide PDB codes and deposition validation reports.

Following this reviewer's suggestion, the corrections to resolution reporting methods are provided in the revised manuscripts (see below).

The cryo-EM 3D maps of Fd-NDH-1L, NDH-1L Δ V and (Fd)-NDH-1L Δ V were deposited in EMDB database with accession codes EMD-0849, EMD-0850 and EMD-0851, respectively. The atomic models of Fd-NDH-1L and NDH-1L Δ V were deposited in PDB with accession codes 6L7O and 6L7P, respectively. The validation reports are provided together with the revised manuscript.

(3) In addition to their structural work, the authors constructed deletion mutants of several of the photosynthetic specific subunits of NDH1. Deletion mutants in most of these subunits

were already examined and most of the results presented in figure 4a are not new, this should be clearly indicated.

Thank for this good point. Following this reviewer's suggestion, we modified the text to clearly cite previous data about the deletion mutants in the revised manuscript as the following (Pages 10, lines 222-232).

“Previous *in vivo* chlorophyll fluorescence studies showed that under growth-light conditions deletion of NdhV or NdhS resulted in decreased NDH-CET activity, whereas removal of NdhO led to increased activity, suggesting that they play different roles in NDH-CET (Battchikova, Wei et al. 2011, Zhao, Gao et al. 2014, Gao, Zhao et al. 2016). To further investigate the roles of OPS regulatory subunits in short-term response to high-light stress, we individually deleted each regulatory subunit in the model cyanobacterium *Synechocystis* sp. strain PCC 6803 (hereafter referred to as *Synechocystis* 6803) and monitored the NDH-CET activity by the *in vivo* chlorophyll fluorescence analysis (Munekage, Hashimoto et al. 2004). Deletion of NdhL almost completely abolished the NDH-CET activity to the same extent as deletion of the essential core subunit NdhI (Fig. 4a). This result was in accordance with our hypothesis that NdhL might play an essential role in stabilizing the PQ-binding cavity (Fig. 2f).”

(4) The authors suggest that NdhV plays a positive role in Fd binding and provide several lines of evidence to support this. The *in-vitro* results generally fall in line with this suggestion, but no evidence *in vivo* is presented.

Thanks for this good point. We agree with the reviewer that the conclusion of the positive regulatory role of NdhV is based on the *in-vitro* experiments and lacks additional *in vivo* evidence. Given that our focus of this paper is the structure of the Fd-NDH-1L complex, we modified the conclusion sentence of this section as “These results strongly support the notion that NdhV likely plays a key role in assisting Fd binding to the NDH-1L complex. Further studies are needed to fully understand the *in vivo* function of NdhV in NDH-CET.” (Page 11, lines 258-260)

(5) With regards to NdhO the authors suggest that it plays a role of negative regulator of NDH1 activity. This set of results is somewhat problematic as the main evidence of the increased activity of NDH1 Δ O comes from a semi quantitative assay (Figure 4E). A more quantitative assay for NDH1 activity done on NDH1 and NDH1 purified from NdhO mutants should be done if the authors insist on including this point.

Following this reviewer's suggestion, we have modified the text to emphasize that our current semi quantitative assay only implies that NdhO might be a negative regulator of NDH-CET activity and future more quantitative studies are required to confirm this hypothesis.

“...these results imply that NdhO might be a negative regulator of NDH-CET activity. Future more quantitative studies are required to confirm this hypothesis and fully understand the regulatory function of NdhO in NDH-CET.” (Page 12, lines 281-283)

We rewrote the section of “Regulatory mechanism of NDH-CET under long-term high-light stress” and removed all the sentences about the function of NdhO (Pages 12-13, lines 285-307).

We removed all the discussions about NdhO in “A model of NDH-CET regulation by high-light exposure” in the Discussion section (Pages 15-16, lines 363-380). Accordingly, we also removed the “red stop sign” that indicates the putative negative role of NdhO in Figure 6 (see revised Fig. 6).

(6) The manuscript appears to be hastily written and in its present form and is difficult to follow.

Thanks for pointing out the language problem. We have carefully rewritten the manuscript and it is much better than the initial version.

(7) Supplementary figure 1E should contain FSC curves for phase randomized volumes as well as masked and unmasked volume and the identify of each curve should be clearly visible. The correlation between the map and model should also be reported on the same graph.

Thanks for this good point. Following this reviewer’s suggestion, we have corrected the FSC curves in the revised manuscript (revised Supplementary Figure 1E).

(8) Supplementary figure 3F should contain FSC curves for phase randomized volumes as well as masked and unmasked volume and the identify of each curve should be clearly visible. The correlation between the map and model should also be reported on the same graph.

Thanks for this good point. Following this reviewer’s suggestion, we have corrected the FSC curves in the revised manuscript (revised Supplementary Figure 3F).

(9) Supplementary figure 10D should contain FSC curves for phase randomized volumes as well as masked and unmasked volume and the identify of each curve should be clearly visible. The correlation between the map and model should also be reported on the same graph.

Following this reviewer’s suggestion, we have corrected the FSC curves and added relevant information (see updated Supplementary Figure 10D).

Minor points –

(10) Line 67 – “Fd-N6a-N6b-N2-PQ” is used without indicating abbreviation of the iron-sulfur clusters.

Following this reviewer’s suggestion, we define the abbreviation of the Fe-S clusters in the revised manuscript as the following.

“...electrons donated by photoreduced Fd are transferred to PQ via the chain of three conserved [4Fe-4S] clusters (Battchikova, Wei et al. 2011, Schuller, Birrell et al. 2019), corresponding to the previously identified clusters N6a, N6b, and N2 in respiratory complex I (Sazanov and Hinchliffe 2006) (Fd-N6a-N6b-N2-PQ) ...” (Page 4, line 67 - 70)

(11) Page 3, line 49 – “improves the Calvin-Benson cycle” seems like the wrong way of stating the better fit between the ATP/NADPH requirements of the CBB cycle and the output of the light reactions.

Thanks for the good point. We rewrote this sentence as “...cyclic electron transfer around photosystem I (PSI CET) is an important mechanism that balances the ATP/NADPH ratio required for the Calvin-Benson cycle...” (See Page 3, lines 48 to 49)

(12) Page 4, line 97 – “Consistent with the recently reported structures of NDH-1L Δ V” should be changed to “recently reported structures of NDH-1”. As it is the authors choice to name this complex NDH-1L Δ V. Same is true for other places in the text where this is done.

Following this review’s suggestion, we changed “NDH-1L Δ V” to “NDH-1” when we cited the recently reported structures in the revised manuscript (Page 3, line 60; page 5, line 100 and page 6, line 121).

(13) Page 5, line 106 – “deactive state” should be “inactive state”

Thanks for this point. Disorder of the β 1- β 2 loop is the key structural feature of “deactive state” of mammalian respiratory complex I (Agip, Blaza et al. 2018, Blaza, Vinothkumar et al. 2018, Letts, Fiedorczuk et al. 2019). Since the β 1- β 2 loop is well ordered in the ubiquinone-binding channel observed in our structure and closely resembles that in active respiratory complex I structures (PDB 4HEA, 6G2J, 5LC5 and 6QBX) (Baradaran, Berrisford et al. 2013, Agip, Blaza et al. 2018, Blaza, Vinothkumar et al. 2018, Letts, Fiedorczuk et al. 2019), we conclude that the structure of the Fd-NDH-1L complex captures an active conformation of NDH-1L.

We have modified the text in the revised manuscript as “Notably, at the proximal end of this cavity, the β 1- β 2 loop of NdhH is well ordered and its conformation closely resembles that in the active state of respiratory complex I structures (Supplementary Fig. 7a,b) (Baradaran, Berrisford et al. 2013, Agip, Blaza et al. 2018, Blaza, Vinothkumar et al. 2018, Letts, Fiedorczuk et al. 2019), suggesting that the structure of the Fd-NDH-1L complex captures an active conformation of NDH-1L” (Page 5, line 108-111).

(14) Page 7, line 153 – “Strikingly, a continuous grease belt,” should be replaced with a more appropriate term.

We have changed “a continuous grease belt,” to “a continuous cofactor belt” in the revised manuscript (page 7, line 160).

(15) Page 11, line 252 - “It has been controversial” should be corrected.

We have rewritten this part and removed this sentence in the revised manuscript.

(16) Line 342 – “Markedly, we found that the negative regulatory OPS subunit NdhO is absent in the NDH-1MS complex (Fig. 6).” - this is not shown in Fig. 6.

Thanks. We have modified the text and removed this sentence in the revised manuscript. We also corrected the figure number mistake (changing “Fig. 6” to “Fig. 5d”) in the revised manuscript (page 15, line 343).

Reviewer #2:

The manuscript by Chunli Zhang et al “Structural insights into NDH-1 mediated cyclic electron transfer” describes the structure of cyanobacterial NDH complex with bound Fd. This is an important study which describes how exactly Fd interacts with NDH complex and how these interactions are regulated under different conditions by expression of specific subunits. The manuscript is generally well written and clearly illustrated, but will benefit from corrections by native English speaker. It will be of interest to the wide Nature Communications readership.

Thanks for pointing out the language problem. We have carefully edited the manuscript and it is much better than the initial version.

Some comments to address in revision:

(1) Line 66 - Which four OPS regulatory subunits?

Following this reviewer’s suggestion, we have added the names of four OPS regulatory subunits (NdhL, NdhO, NdhS and NdhV) in the revised manuscript (see page 4, lines 72).

(2) 106 – Structure of ovine complex I (5LNK) is not of deactive state, but of the “open” state, likely part of the catalytic cycle along with the “closed” state (see e.g. pubmed 31492636). In the true deactive state β 1- β 2 loop is disordered.

Thanks for pointing out this mistake. Following this reviewer’s suggestion, we removed the comparison between NDH-1L and ovine complex I (5LNK) in revised Supplementary Figure 7.

Disorder of the β 1- β 2 loop is the key structural feature of “deactive state” of respiratory complex I (Agip, Blaza et al. 2018, Blaza, Vinothkumar et al. 2018, Letts, Fiedorczuk et al. 2019). Since the β 1- β 2 loop is well ordered in the ubiquinone-binding channel observed in our structure and it closely resembles that in the active respiratory complex I structures (PDB 4HEA, 6G2J, 5LC5 and 6QBX) (Baradaran, Berrisford et al. 2013, Agip, Blaza et al. 2018, Blaza, Vinothkumar et al. 2018, Letts, Fiedorczuk et al. 2019), we conclude that the structure of the Fd-NDH-1L complex captures an active conformation of NDH-1L.

We have modified the text in the revised manuscript as “Notably, at the proximal end of this cavity, the β 1- β 2 loop of NdhH is well ordered and its conformation closely resembles that in the active state of respiratory complex I structures (Supplementary Fig. 7a,b) (Baradaran, Berrisford et al. 2013, Agip, Blaza et al. 2018, Blaza, Vinothkumar et al. 2018, Letts, Fiedorczuk et al. 2019), suggesting that the structure of the Fd-NDH-1L complex captures an active conformation of NDH-1L” (Page 5, line 108-111).

(3) 128 – should be discussed how exactly β -carotene molecules stabilize the complex (contacts, environment).

Thanks for this good point. Following this reviewer's suggestion, we have modified this section extensively to include discussion of how β -carotene molecules stabilize the complex. We also added a new figure in supplementary materials (Page 7 lines 143-156, Page 8 lines 168-175, and revised Figure 2 and Supplementary Figure 9).

(4) 296-299 – very convoluted sentence, should be re-phrased.

Following this reviewer's suggestion, in the revised manuscript we removed that sentence and rewrote the section in a clear and concise manner to emphasize the conclusion “the NDH-1MS complex without NdhO is induced to accelerate the NDH-CET activity by long-term high-light irradiation” (Page 13, lines 305 to 307).

(5) 326-329 – This speculation seems unlikely, there are plenty of Fe-containing cofactors in cyanobacteria/chloroplast enzymes. The short cluster chain is simply inherited from hydrogenases.

Thanks for this good point. We have modified the sentence as the following.

“As a consequence, the photosynthetic NDH-1L complex had developed a shorter electron transfer chain Fd-N6a-N6b-N2-PQ, which was inherited from its [NiFe] hydrogenase ancestor (Peltier, Aro et al. 2016).” (Page 14, lines 332-334)

(6) 363-368 – The difference between growth and high-light short term is not clearly stated or shown in figure – presumably NdhV expression levels are much higher in the latter case?

Thanks for this good point. We agree with the reviewer that the difference between growth and high-light short term is not clearly stated. Within 5 minutes (short-term response), the total amount of NdhV under high light exposure is similar to that under growth light condition (Figure 5a shows that within 5 min of high light exposure the mRNA level of NdhV should be similar to that under growth light condition). We propose that under growth light conditions, the low level of photoreduced Fd from PSI only needs a fraction of NdhV to assist its binding to NDH-1L. We rewrote the discussion to clearly state the difference between growth light and short-term high-light exposure in the revised manuscript.

“We propose that, under growth light conditions, the low level of photoreduced Fd from PSI only needs a fraction of NdhV to assist its binding to the constitutively expressed NDH-1L, restraining the electron transfer to balance the ATP/NADPH ratio required by Calvin-Benson cycle (Fig. 6). When cyanobacteria are challenged by a short-term high-light exposure, we propose that, although the expression level of NdhV remains largely unchanged (Figure 5a), the available NdhV molecules are sufficient to associate with increased photoreduced Fd to bind to the NDH-1L complex and to accelerate the electron transfer rate for the need of efficient photosynthesis (Fig. 6)”. (Page 16, lines 366-373)

References:

- Agip, A. A., J. N. Blaza, H. R. Bridges, C. Viscomi, S. Rawson, S. P. Muench and J. Hirst (2018). "Cryo-EM structures of complex I from mouse heart mitochondria in two biochemically defined states." Nat Struct Mol Biol **25**(7): 548-556.
- Baradaran, R., J. M. Berrisford, G. S. Minhas and L. A. Sazanov (2013). "Crystal structure of the entire respiratory complex I." Nature **494**(7438): 443-448.
- Battchikova, N., L. Wei, L. Du, L. Bersanini, E. M. Aro and W. Ma (2011). "Identification of novel Ssl0352 protein (NdhS), essential for efficient operation of cyclic electron transport around photosystem I, in NADPH:plastoquinone oxidoreductase (NDH-1) complexes of *Synechocystis* sp. PCC 6803." J Biol Chem **286**(42): 36992-37001.
- Blaza, J. N., K. R. Vinothkumar and J. Hirst (2018). "Structure of the Deactive State of Mammalian Respiratory Complex I." Structure **26**(2): 312-319 e313.
- Gao, F., J. Zhao, X. Wang, S. Qin, L. Wei and W. Ma (2016). "NdhV Is a Subunit of NADPH Dehydrogenase Essential for Cyclic Electron Transport in *Synechocystis* sp. Strain PCC 6803." Plant Physiol **170**(2): 752-760.
- Laughlin, T. G., A. N. Bayne, J. F. Trempe, D. F. Savage and K. M. Davies (2019). "Structure of the complex I-like molecule NDH of oxygenic photosynthesis." Nature **566**(7744): 411-414.
- Letts, J. A., K. Fiedorczuk, G. Degliesposti, M. Skehel and L. A. Sazanov (2019). "Structures of Respiratory Supercomplex I+III2 Reveal Functional and Conformational Crosstalk." Molecular Cell **75**(6): 1131-1146.e1136.
- Munekage, Y., M. Hashimoto, C. Miyake, K. Tomizawa, T. Endo, M. Tasaka and T. Shikanai (2004). "Cyclic electron flow around photosystem I is essential for photosynthesis." Nature **429**(6991): 579-582.
- Peltier, G., E. M. Aro and T. Shikanai (2016). "NDH-1 and NDH-2 Plastoquinone Reductases in Oxygenic Photosynthesis." Annu Rev Plant Biol **67**: 55-80.
- Sazanov, L. A. and P. Hinchliffe (2006). "Structure of the hydrophilic domain of respiratory complex I from *Thermus thermophilus*." Science **311**(5766): 1430-1436.
- Schuller, J. M., J. A. Birrell, H. Tanaka, T. Konuma, H. Wulfhorst, N. Cox, S. K. Schuller, J. Thiemann, W. Lubitz, P. Setif, T. Ikegami, B. D. Engel, G. Kurisu and M. M. Nowaczyk (2019). "Structural adaptations of photosynthetic complex I enable ferredoxin-dependent electron transfer." Science **363**(6424): 257-260.
- Zhao, J., F. Gao, J. Zhang, T. Ogawa and W. Ma (2014). "NdhO, a subunit of NADPH dehydrogenase, destabilizes medium size complex of the enzyme in *Synechocystis* sp. strain PCC 6803." J Biol Chem **289**(39): 26669-26676.

Reviewers' Comments:

Reviewer #1:

Remarks to the Author:

The current version of the manuscript answers my concerns and I recommend that it will be accepted for publication.

Reviewer #2:

Remarks to the Author:

The manuscript has been revised adequately and could be accepted, except that provided PDB validation report appears to show that submitted map was not aligned with the model, and so all statistics on map-model fit are very bad. Authors should re-submit the map with fitted model and provide updated validation report before the manuscript can be accepted.

Response to Reviewers' comments (NCOMMS-19-32222A)

Reviewer #1:

The current version of the manuscript answers my concerns and I recommend that it will be accepted for publication.

Thanks.

Reviewer #2:

The manuscript has been revised adequately and could be accepted, except that provided PDB validation report appears to show that submitted map was not aligned with the model, and so all statistics on map-model fit are very bad. Authors should re-submit the map with fitted model and provide updated validation report before the manuscript can be accepted.

Thanks for pointing this problem. To find the reasons that caused the problem (the submitted map was not aligned with the model), we first checked the original coordinate and map files and found no mismatch. Then we downloaded the coordinate and map files that have been processed by RCSB and compared them with the original files. The coordinate files are completely identical, so the processed coordinates (pink) and the original maps (grey) aligned well (Figure 1). However, some apparent shift could be traced between the processed maps (yellow) and the original maps (grey) (Figure 2), suggesting that this shift of the maps resulted in the problem that the processed maps (yellow) could not be aligned with the processed coordinates (pink) (Figure 3).

To avoid the unexpected effects of the processed map and coordinate files in old depositions, we resubmitted the original coordinate and map files in new depositions and the new validation reports are included in our resubmission of the paper.

Figure 1. Comparison of the processed coordinates (pink) and original maps (grey). Upper: Fd-NDH-1L; Lower: NDH-1LAV

Figure 2. Comparison of the processed maps (yellow) and original maps (grey). Upper: Fd-NDH-1L; Lower: NDH-1LΔV

Figure 3. Comparison of the processed coordinates (pink) and processed maps (yellow).
Upper: Fd-NDH-1L; Lower: NDH-1LAV